# Interacting Networks of the Hypothalamic–Pituitary–Ovarian Axis Regulate Layer Hens Performance

**DOI:** 10.3390/genes14010141

**Published:** 2023-01-04

**Authors:** Jinbo Zhao, Hongbin Pan, Yong Liu, Yang He, Hongmei Shi, Changrong Ge

**Affiliations:** 1Faculty of Animal Science and Technology, Yunnan Agricultural University Kunming, Kunming 650201, China; 2Branch of Animal Husbandry and Veterinary of Heilongjiang Academy of Agricultural Sciences, Qiqihar 161005, China

**Keywords:** hypothalamic–pituitary–ovarian axis, laying performance, molecular mechanism, interactive network regulation mechanism

## Abstract

Egg production is a vital biological and economic trait for poultry breeding. The ‘hypothalamic–pituitary–ovarian (HPO) axis’ determines the egg production, which affects the layer hens industry income. At the organism level, the HPO axis is influenced by the factors related to metabolic and nutritional status, environment, and genetics, whereas at the cellular and molecular levels, the HPO axis is influenced by the factors related to endocrine and metabolic regulation, cytokines, key genes, signaling pathways, post-transcriptional processing, and epigenetic modifications. MiRNAs and lncRNAs play a critical role in follicle selection and development, atresia, and ovulation in layer hens; in particular, miRNA is known to affect the development and atresia of follicles by regulating apoptosis and autophagy of granulosa cells. The current review elaborates on the regulation of the HPO axis and its role in the laying performance of hens at the organism, cellular, and molecular levels. In addition, this review provides an overview of the interactive network regulation mechanism of the HPO axis in layer hens, as well as comprehensive knowledge for successfully utilizing their genetic resources.

## 1. Introduction

Egg production is one of the vital biological and economic traits for poultry breeding, which has profound economic impacts on the laying hen industry. The concept, “700 eggs produced in 500 days”, proposed by the French company Institute Selection Animal in 2011 [1], provides a clear goal for the China layer hens breeding industry. This concept has also been included in the future goals of the China Layer Hens Genetic Improvement Plan (2021–2035) [2]. The current production level of layer hens in China is 14–18 kg egg mass per hen, and a few domestic laying farms have achieved 20 kg per hen. However, internationally, the production level of layer hens has reached 20–21 kg per hen, indicating the gap in layer hens performance in China.

Over the last 50 years, layer hens’ egg production has been influenced by different factors such as genetics (40%), nutrition (20%), disease control (15%), feeding management (20%), and environment (5%) (Figure 1) [3], and these factors directly or indirectly affect the egg production of layer hens. It is commonly believed that egg production is directly regulated by the hypothalamus–pituitary–ovarian (HPO) axis, which comprises a comprehensive network of regulatory molecules, such as endocrine hormones and metabolic cytokines, and these molecules activate or inhibit the HPO axis signaling pathways. Furthermore, the influence of methylation or epigenetic modifications is a biological characteristic that has an economic impact on the layer hen breeding industry [4,5,6,7,8,9]. Noncoding RNAs also regulate the target genes related to endocrine and metabolism signaling pathways. A study reported that microRNAs present in avian ovarian tissue, such as gga-miR-34b, gga-miR-34c, and gga-miR-216b, were differently expressed in high-yield laying hens [10]. By binding to the target genes, noncoding RNAs regulate the synthesis, transport, and secretion of genes related to reproductive hormones downstream of signaling pathways, namely *STAR*, *CYP11A1*, and *CYP19A1* [11,12].

The present review focuses on interacting networks of the HPO axis regulating layer hen performance through endocrine and metabolic regulations, providing an overview of the interactive network regulation mechanism of the HPO axis in layer hens and comprehensive knowledge for successfully utilizing their genetic resources.

## 2. Organism-Level Regulation Mechanism of the HPO Axis

### 2.1. Development Characteristics of the HPO Axis

The hypothalamus, the central regulator of physiological homeostasis, regulates reproductive activities in hens [13]. The hypothalamus, pituitary gland, and ovary process signals through positive and negative feedback precision regulation. The hypothalamus secretes gonadotropin-releasing hormone (GnRH) or gonadotropin inhibitory hormone (GnIH) through physiological stimulation, controlling the secretion of anterior pituitary prolactin (PRL), follicle-stimulating hormone (FSH) [14], luteinizing hormone (LH) [15], and oxytocin (OXT), which are associated with egg production performance. FSH, LH, and progesterone (P_4_) secretion levels progressively increase at the peak stage of laying, whereas P_4_, estradiol (E_2_), and PRL secretion concentrations gradually increase at the time of the first egg. The cascade response triggered by GnRH and GnRHR regulates the HPO axis. Research has shown that distinct GnRH might have different receptor types. GnRH is classified into two major classes in most vertebrates, namely GnRH-I and GnRH-II. GnRH-I operates primarily on the pituitary gland, whereas avian GnRH, which is evolutionarily highly conserved, is related to GnRH-II [16,17]. GnRH-I and GnRH-II signals produced by the hypothalamus are significantly correlated with egg production and body weight of White Leghorn laying hens. However, unlike GnRH gene expression, *GnIHR* mRNA expression in the hypothalamus is significantly higher in the pituitary of sexually immature chicken than in the pituitary of mature chicken. When avians achieve sexual maturity and begin to produce gonadotropins, a surge in circulating E_2_ and progesterone concentrations may downregulate *GnIHR* gene expression. Furthermore, GnIHR protein expressed in *FSHβ* mRNA- or *LHβ* mRNA-containing cells is likely to promote GnIH-mediated inhibition of LH and FSH secretion. *BMP15* and *FSHR* genes are also linked to reproduction and hence to the laying performance [18]. Therefore, GnRH may play an important role in the initiation signal of the physiological process of egg production. Furthermore, vasoactive intestinal peptides (*VIPs*) secreted by the hypothalamus inhibit GnRH secretion through its receptor (*YIR*) and regulate ovulation [19]. The positive and negative feedback regulation of these hormones and neuroendocrine pathways are crucial for avian reproductive performance. Hence, the primary challenge in improving the reproductive performance of laying hens is to elucidate the key mechanisms controlling the HPO axis and identifying the differently expressed genes involved.

The development of chicken ovaries is regulated by hormones secreted in the hypothalamus and the pituitary gland, Ovarian development is the key element determining egg production of hens [20]. The ovary of a chicken begins to grow at approximately 14 weeks and attains sexual maturity between the age of 18 and 20 weeks [21,22]. During hens’ reproductive life, most ovarian follicles undergo atresia, with only approximately 5% progressing to the final hierarchical stages of maturation and ovulation. Growth of the avian ovarian follicle occurs in two stages: the pre-hierarchical follicle stage and the hierarchical follicle stage [23,24]. Pre-hierarchical follicles have been classified as small white follicles (SWF, 2–4 mm), large white follicles (LWF, 4–6 mm), and small yellow follicles (SYF, 6–8 mm), whereas hierarchical follicles are the pre-ovulation follicles (9–12 mm) that have been classified using a rigorous grading system from grade F6 to grade F1, with F1 representing mature follicles and F6 representing hierarchical follicles destined to become fully mature follicles. This is a highly hierarchical process in which one follicle is picked from the small yellow follicles pool and placed in the hierarchical follicle pool sequential order, which then grows into ovulatory follicles [25,26,27,28].

In addition to analyzing the level of egg-laying performance from the perspective of hormones, the challenges related to the bird’s phenotype [29] involve commonly used qualities, such as age at first egg (AFE) [30], egg weight, egg number at the age of 300 days (EN300), and other external indicators used for artificial selection to improve egg production efficiency [31].

Figure 2a shows that egg production of layer hens is regulated by the HPO axis. The hypothalamus receives activating signals and secretes the gonadotropin-releasing hormone (GnRH) or gonadotropin inhibitory hormone (GnIH). The pituitary gland receives signals from the hypothalamus, which binds to *GnRHR* receptors on the pituitary gland and secretes anterior pituitary prolactin (PRL), FSH, and LH. The ovary receives signals from the pituitary gland and secretes progesterone (P_4_) and estradiol (E_2_), both of which play a vital role in regulation by initiating follicle selection. Figure 2b shows the mechanisms of steroid hormone production by granulosa and theca cells. Steroid hormones are synthesized via two main pathways; granulosa cells (GCs) regulate the synthesis and secretion of P4 through the transcription of *STAR* and *CYP11A1* genes, whereas theca cells regulate the synthesis and secretion of E2 through *17βHSD* and *CYP19A1* gene transcription (Figure 2c). These genes involved in reproductive hormone synthesis are transported to the nucleus and exert the biological effect of follicle selection by activating the nuclear receptor signaling pathway.

### 2.2. Factors Affecting the Organism-Level Regulation Mechanism

The layer hens performance is not only regulated by HPO axis, but also affected by genetic, environmental, and nutritional metabolic status.

#### 2.2.1. Factors Related to the Metabolic and Nutritional Status

Nutrition affects the secretion of reproductive hormones and the concentration of metabolic substrates in blood. Research showed that food restriction promoted release of P4, reduced out of the T, but did not affect E release, suggesting that metabolic status can control the release of these hormones [32]. The estrogen estradiol-17ß is known as one of the major gonadal steroid hormones with different functions in reproduction [33]. Modern layer hens have consistently high plasma estradiol-17ß levels [34], estrogen divided into two types of receptors, ER receptor α and ER receptor β. The mRNA expression of Erα was significantly higher compared to *ERβ* in layer hens granulosa cells [35], suggesting that estrogen plays a role in regulating follicular development through ERα receptors. Previous research reported that follicle selection is indispensable in the reproductive process in female chickens, the symbol of which is the proliferation and differentiation of granulosa cells (GCs), In chickens follicular GCs, the expression of steroidogenic acute regulatory protein (*StAR*) and cytochrome P450 family 11 subfamily A member 1 (*CYP11A1*) is a prerequisite for progesterone synthesis and is related to follicle selection [36,37,38]. It suggests that estrogen and progesterone work together to complete the physiological process of follicle selection in layer hens. Many studies have shown that estrogen is critical for lipid metabolism in the liver. In vivo and in vitro experiments showed that estrogen decreased the mRNA expression of *PPARGC1B*, which had been reported to be linked with lipid metabolism, by directly increasing the expression of *miR-144-3p* [39,40]. These findings show that the liver ERα controls the regulation of liver lipid metabolism, thus connecting liver lipid metabolism to the reproduction cycle.

#### 2.2.2. Environmental Factors

Environmental factors exert a substantial impact on the HPO axis and thus can be utilized as exogenous factors to directly regulate the laying performance. Light and temperature are the key environmental factors. Firstly, light affects the hypothalamus, stimulating GnRH neurons to produce GnRH-II in response to light cues and regulating the reproduction cycle by modulating melatonin production [41]. Secondly, the temperature affects the laying performance of hens in two ways. Research has shown that cold stress can increase free radicals in the body, which affects the antioxidant capacity and causes an imbalance in the stable state of free radicals, thus representing the main three-way linkage reaction of oxidative stress, inflammatory response, and immunological stress [42]. A change in temperature of the environment triggers a series of neuroendocrine responses, causing a systemic control reaction to adjust the endocrine system. To maintain a constant body temperature, the HPO axis is activated, which accelerates metabolism, enhances protein breakdown, and increases insulin secretion [43]. 

#### 2.2.3. Genetic Factors

Heredity factors also play a crucial role in the hens’ egg-laying performance. Differentially expressed genes (DEGs) were identify in the hypothalamus and pituitary tissues of high and low-yield hens by transcriptomics. Laying performance is linked to the signaling pathways, such as extracellular matrix, glycosaminoglycan biosynthesis, protein extracellular matrix, and extracellular space. GWAS was used to mine SNP loci in commercial egg-type chickens and indigenous chickens. Some genes, such as *NELL2* (neural EGFL like 2), *KITLG* (KIT ligand), *GHRHR* (Growth hormone releasing hormone receptor), *NCOA1* (Nuclear receptor coactivator 1), *ITPR1* (inositol 1, 4, 5-trisphosphate receptor type 1), *GAMT* (guanidinoacetate N-methyltransferase), and *CAMK4* (calcium/calmodulin-dependent protein kinase IV), deserve our attention and further study since they have been reported to be closely related to egg production, egg number, and reproductive traits. In addition, the most significant genomic region obtained in this study was located at 48.61–48.84 Mb on GGA5. In this region, we have repeatedly identified four genes, in which *YY1* (YY1 transcription factor) and *WDR25* (WD repeat domain 25) have been shown to be related to oocytes and reproductive tissues, respectively, which implies that this region may be a candidate region underlying egg number traits [44]. Hence, GWAS and transcriptome analyses and other modern technologies need to be applied to investigate the genes related to laying traits. Such investigation might accelerate the improvement of existing genotypes and the breeding of new breeds.

## 3. Cellular-Level Regulation of the HPO Axis in Avians

### 3.1. Insulin-like Growth Factor

Because of the complicated physiological effects of insulin-like growth factors and growth hormone, the roles of growth hormone and IGF-I are often concurrently investigated. Growth hormone (GH) and IGF-I exert biological effects by attaching to their receptors and engaging in downstream signaling cascades. Autocrine production of insulin-like growth factor-I mediates growth hormone-mediated DNA synthesis and proliferation in primary cultured hepatocytes of adult rats. GH is expressed in the GCs of avian ovarian follicles, suggesting that GH is involved in the development of the avian reproductive system via autocrine/paracrine secretion [45]. In vitro tests on primary cultured cells have revealed that GH enhanced the autocrine capacity of IGF-I via the JAK2/PLC signaling pathway of its receptor [46]. Subsequently, its receptor activated the tyrosine kinase signaling pathway of the IGF-I receptor, resulting in the biological impact of IGF-I [46]. This result indicates that growth hormone (GH) and insulin-like growth factor (IGF-1) are fundamental in poultry follicle development.

### 3.2. Epidermal Growth Factor

Pituitary gonadotropins (FSH and LH) play a vital role in ovarian development. However, pieces of evidence suggest that the local growth-promoting factors may influence gonadotropin secretion and enhance oocyte maturation through paracrine and autocrine mechanisms. Epidermal growth factor (EGF) interacts with the KIT receptor of oocytes via its ligand, thereby triggering the recruitment of primordial follicles and boosting oocyte development. EGF, transforming growth factor α (TGF-α), heparin-binding EGF-like growth factor (HB-EGF), amphiregulin (AR), epiregulin (ER), betacellulin (BTC), and β cytokine, are members of the EGFR family [47]. AR, ER, and BTC are expressed in the granulosa and thecal layers of ovarian follicles [48]. Exogenous EGF increases GC proliferation, inhibits granulosa cell apoptosis, and regulates steroid hormone production by GCs [49]. These findings suggest that heparin-binding EGF-like growth factor (HB-EGF) in the avian ovary can be utilized as an oocyte signal source to regulate granulosa cell proliferation.

### 3.3. Transforming Growth Factor β

TGF-β is a versatile cytokine of the transforming growth factor β superfamily that is primarily involved in the formation of extracellular matrix (ECM) in poultry. TGF-β is involved in the primary follicular development signaling pathway that controls follicular development [50]. The TGF-β signaling pathway is crucial in selecting hierarchical follicles from the small yellow follicle pool. The physiological process of follicular selection is affected by many endocrine, autocrine, and paracrine pathways. The major component of ECM is collagen, which is mostly dispersed in the follicular membrane layer and generated with the development of follicular GCs and follicular expansion [51]. These results indicate that ECM can bind to TGFβ, activate the TGFβ signaling pathway and regulate the process of follicle selection.

### 3.4. Immune Regulatory Factors

The occurrence of reproductive disorders in poultry is strongly correlated with the HPO axis and neuro–endocrine–immune network molecules, such as tumor necrosis factor α (TNF-α) and interferon γ (IFN-γ), represented by inflammatory cytokines such as interleukin-10 (IL-10) and interleukin-6 (IL-6). The hosts’ natural immunity maintains follicular development, ovulation, fertilization, and egg formation in poultry, which may be related to several neurotransmitter receptors and endocrine hormone receptors on the surface of proliferating immune cells. Cell surface receptors and their ligands promote or inhibit immune cells. The nervous, endocrine, and immune systems interact to produce neurotransmitters, hormones, and cytokines, thereby forming a nerve–endocrine–immune network control system to maintain homeostasis in the body [52]. These results show that immune factors may play an important role in layer hens reproductive performance.

## 4. Molecular Regulation of the HPO Axis in Avians

### 4.1. The Role of Reproduction-Related Genes in HPO Axis Development

Transcriptome technology has been utilized to identify differentially expressed genes of the HPO axis, and the KEGG enrichment pathway analysis has been employed to establish the HPO signaling pathways. A study identified 414 differentially expressed genes in the pituitary gland, enriched in 108 GO terms in the high-laying and low-laying groups, which were connected to 4 enrichment signaling pathways. The ovary was found to have 356 differentially expressed genes enriched in 4 signaling networks, whereas the hypothalamus exhibited no KEGG signaling pathway [53]. Research has shown that GEGs were identified and analyzed in the HPO axis by transcriptome, and these genes are associated with layer hens reproductive performance. Rho-associated coiled-coil containing protein kinase 2 (*ROCK2*) is involved in the regulation of GTP synthesis in the hypothalamus, which may affect GnRH secretion after being activated by signals and directly interact with the pituitary gland to secrete gonadotropin, thereby acting on target organs and participating in biological activities [22]. Insulin-like growth factor binding protein 7 (*IGFBP7*) is an inducer of ovarian follicular development that is involved in the ovarian response to FSH [54]. The GTPase activating the Rap/Ran GAP domain-like 1 gene (*GARNL1*) is linked to EN300 and AFE [19].The POU class 1 homeobox 1 gene (*POU1F1*) is primarily responsible for activating promoters of the genes producing GH, prolactin, and the gonadotropin chain [55].

Most of the genes found are involved in follicles development, such as bone morphogenetic protein 15 (*BMP15*), which is a growth factor produced by oocytes involved in mammalian ovarian development and ovulation [56]. Steroidogenic acute regulatory protein (*StAR*) is a fast steroid synthesis regulatory protein that acts along with the cytochrome P450 family 11 subfamilies A member 1 (*CYP11A1*) gene to complete follicular selection [57,58]. *CYP11A1*, *CYP17A1*, and 17β-hydroxysteroid dehydrogenase type 2 (*17β-HSD2*) control the production of steroid hormones, including estrogen and progesterone, and among these genes, the expression of *CYP11A1* is the most significant [59]. *CYP11A1* expression in avian follicular GCs is required for progesterone secretion [60]. The major components of ECM, namely collagen type-VI α 2 chain (*COL6A2*), collagen type-IV α 1 chain (*COL4A1*), collagen type-IV α 2 chain (*COL4A2*), and collagen type-VI α 1 chain (*COL6A1*) are involved in the production of steroid hormones. ECM is primarily composed of collagen [61], laminin, proteoglycan, fibronectin, and glycoprotein, which influence biological processes, such as cell differentiation, proliferation, adhesion, morphogenesis, and phenotypic expression [62,63,64]. In several species, IGF-I enhances oocyte maturation. Bone morphogenetic protein 6 (*BMP6*) can increase FSH and anti-Mullerian hormone (AMH) expression in GCs, and AMH can promote the differentiation of GCs [65]. Inhibin subunit β B (*INHBB*) genes can induce early differentiation of GCs and enhance their capacity to stimulate steroid production. The function of these genes also suggests that they are involved in follicle selection in chickens.

### 4.2. The Role of Signaling Pathways in Granulosa Cell Development

The HPO axis molecular signaling pathways play a crucial role in the laying performance of hens [66]. The whole-genome and transcriptome analysis techniques have been extensively applied in studying avian skeletal muscle development [67], examining sperm movement regulating factors [68], diagnosing poultry disease [69], and understanding genetic diversity [70]. However, the genes regulating egg-laying performance on the HPO axis are still poorly studied. Research has shown that the HPO axis is abundant in several signaling pathways associated with follicle selection. The pathway plays an important role in granulosa cell proliferation, differentiation, and apoptosis, including the mTOR signaling pathway, Wnt signaling pathway, TGF-β signaling pathway, ECM receptor signaling pathway, and steroid biosynthesis signaling pathway (Figure 3).

#### 4.2.1. Wnt Signaling Pathway

The Wnt signaling pathway plays a crucial role in the follicle selection of hens [71]. This pathway can regulate granulosa cell proliferation in follicles to increase steroid hormone production, and the biological mechanism of follicle selection in hens depends on the proliferation and differentiation of GCs [71]. The ovulation process in hens is hierarchical and begins from F6 to F1 follicles [25]. It is estimated that the ovary of sexually mature hens contains approximately 12,000 oocytes [28]; however, only a few of these oocytes reach the ovulation stage, and most small yellow follicles are not selected and undergo apoptosis [25]. Research has shown that *BMPs, AHM, FSH, OCLN, STAR, CPY11A1*, and other genes are expressed in ovarian GCs. *OCLN* and *STAR* have been linked to follicle selection; these genes may mediate the activation of a β-catenin-dependent pathway in the Wnt signaling pathway, which regulates GC proliferation, differentiation, and apoptosis. In the Wnt4 signaling pathway, wnt4 mRNA expression peaks at the small yellow follicles stage that promotes the proliferation of GCs [71]. The mRNA expression levels of *STAR* and *CYP11A1* were enhanced in the pre-hierarchical follicle and hierarchical follicle stages. The Wnt signaling pathway promotes follicle selection and collagen synthesis in the ECM. Hence, the Wnt signaling pathway is a key biological signaling route in the follicular selection process. Further research is required to explore the molecular mechanism of the Wnt signaling pathway in improving the degree of follicle development and decreasing the number of follicles atretic. This could lay the groundwork for identifying molecular genetic markers of laying performance in hens and breeding new breeds.

#### 4.2.2. mTOR Signaling Pathway

The mTOR signaling pathway is a conserved serine/threonine-protein kinase pathway that integrates extracellular signals and affects gene transcription and protein translation by phosphorylating the downstream target protein ribosomal P70S6 kinase, thereby regulating the biological processes, such as cell growth, proliferation, and differentiation [72,73]. Abnormal mTOR signaling pathways can cause reproductive disorders, such as premature ovarian failure and polycystic ovarian syndrome (PCOS) [74], cancer [75], diabetes [76], neurological disorders [77], and aging [78]. According to research, the mTOR signaling pathway regulates follicle development, meiosis and oocyte maturation, ovarian cell differentiation and steroid hormone release, sexual maturity, ovarian aging, and embryo development. During ovary development, the mTOR signaling pathways play a vital role in the development of the avian ovary and function in two ways. First, the pathway might be involved in the primordial follicle activation, promote sexual maturity, determine the ovarian reserves, and influence the length of the reproductive cycle, thus playing a vital role in avian ovary development. Second, the pathway mediates granulosa cell sensitivity to FSH and promotes granulosa cell growth. It primarily mediates the cyclic adenosine monophosphate/protein kinase A system signaling pathway (*cAMP*/*PKA*), which subsequently activates the mTOR signaling pathway and promotes follicle growth [79]. Research has shown that treatment of the 75-week and 35-week chicken granulosa cells with mTOR agonist MHY1485 enhances GCs proliferation and inhibits GCs apoptosis in 75-week layer hens, suggesting that the mTOR signaling pathway can regulate ovarian follicular development in aged laying hens [80].

#### 4.2.3. TGF-β Signaling Pathway

TGF-β functions in tissue growth, homeostasis regulation, and repair [81] and is produced by immune cells [82] including T and B cells, dendritic cells, and macrophages. TGF-β is activated by a complex composed of *LABP* and *LAP*, which phosphorylates the TGF-βR-I receptor, activating *SAMD* and interacting with transcription factors at DNA sequence-specific sites *ATF2* and *SBE* in the promoter region to mediate the TGF-β biological function. This regulates gene expression [83]. Additionally, TGF-β promotes other SAMD signal transduction pathways, including numerous mitogen-activated protein kinase (*MAPK*) and MAPK/ERK kinase (*MEK*) pathways, which are involved in *RhoA* and *Ras* upstream activation [84]. Depending on the cell type, TGF-β produces a range of complicated physiological responses, including late G1 growth arrest, differentiation programmed modifications, and apoptosis involving the phosphatidylinositol-3-kinase (*PI3K*) and protein phosphatase-2A. The differentiation, proliferation, and aging processes of follicles are all crucial for the selection of avian follicles, and the extracellular matrix signaling pathway is also implicated in the selection of avian follicles, resulting in the rapid growth and development of follicles. Research has shown that during follicle growth, collagen was secreted by GCs under TGF-β1 stimulation and was subsequently collaboratively transferred to neighboring TCs to increase cell proliferation and thus to promote follicle development via an intercellular cooperative pattern during development of chicken growing follicles [85]. These result show that TGF-β signaling pathway plays a critical role in follicular selection.

#### 4.2.4. ECM Receptor Signaling Pathway

The ECM signaling pathway, with collagen as the major component, is a key pathway that influences follicle growth and development [86]. Cells modify the ECM through degradation and recombination mechanisms and interact with integrins to govern the cell differentiation and proliferation processes [87]. In addition, ECM can combine with growth factors, such as vascular endothelial growth factor, hepatocyte growth factor, and *BMP*, to regulate germ cell formation [88]. Actin recombination also regulates cell proliferation via the ECM. In this process, aberrant ECM recombination can result in various changes, among which pathological changes disrupt the ECM via the regulatory impact, thus altering cell differentiation, proliferation, and apoptosis. Matrix metalloproteinases (*MMPs*) are involved in several biological processes, such as follicle development, ovulation, atresia, and degeneration, all of which require ECM remodeling. PRL regulates the transcription and translation of certain components of the MMP metalloenzyme protein system, thereby regulating the proliferation of GCs [89].

### 4.3. The Role of Epigenetic Modification in Granulosa Cell Development

Noncoding RNA, DNA methylation, and RNA methylation is an important information carrier in epigenetic modification. With the continuous optimization and improvement of transcriptome technology, increasing noncoding sequences, which were originally considered as “transcriptional noise”, have been discovered [90]. Noncoding RNAs represented by microRNAs, long noncoding RNAs, circRNAs, and PiWi protein-interacting RNAs (piRNAs) play a vital role in granulosa cell differentiation, proliferation, growth, apoptosis, and other important physiological processes [91,92,93]. Therefore, examining the role of noncoding RNAs in GCs is essential.

#### 4.3.1. The Role of Noncoding RNAs in Granulosa Cell Development

MicroRNAs (miRNA) are endogenous noncoding RNAs that exhibit a range of biological activities, including cell proliferation, differentiation, apoptosis, gonad development, and lipid metabolism [94,95]. Under the action of RNA polymerase II, miRNA primarily develops a lengthy stem–loop structure, referred to as primary miRNA (pri-miRNA). Pri-miRNA is processed into a pre-miRNA with a stem–loop structure of approximately 70 nucleotides under the action of the ribozyme Drosha III and other cofactors [96]. Pri-miRNA is subsequently carried to the nucleus by the RNA-GTP enzyme and exportin 5, where it is converted by nuclease into mature miRNAs with a length of approximately 22 nucleotides. Research has revealed that the mature miRNAs are expressed in various tissues, including embryos [97], ovaries, adipose tissues [98], lungs, and immunological organs. The role of miRNAs in avian ovarian development has been extensively studied. Transcriptomic studies of avian ovaries have revealed that the quantity of miR-26a-5p differs significantly between sexually mature and immature chickens [99]. AFE was linked to 10 single nucleotide polymorphisms, and miR-26a-5p reduced the expression of the target gene *TRNCA* [100]. The antiapoptotic gene B-cell lymphoma 2 (*BCL-2*) is upregulated, which regulates granulosa cell growth [100]. Simultaneously, miR-1a and miR-21 are differentially expressed in sexually mature and immature hens and follicles at various developmental stages, suggesting that miRNAs either inhibit mRNA translation or promote mRNA degradation, eventually affecting follicle development in the ovary [101]. The miRNAs are stably present in human and animal serum and plasma [102]. Therefore, it is a novel class of biomarkers for diagnosis of cancer and other diseases. However, a few studies have been conducted on the role of miRNAs in chicken GCs. As shown in Table 1.

Long noncoding RNAs (LncRNA) are longer than 200 nucleotides and lack protein coding functions and are pivotal epigenomic factors [115]. Additionally, they are involved in many biological functions, such as mRNA splicing, gene regulation, and protein stabilization [116,117]. LncRNAs are also essential molecular components that regulate the laying performance and play a role in post-translational mRNA processing [118,119]. Long noncoding RNAs control the formation of small yellow follicles in the ovary [120]. Many distinct mRNAs may play a role in follicle growth by regulating steroid hormone production, oocyte meiosis, and the p13K-Akt signaling pathway [121]. A transcriptomic-based analysis identified 160 mRNAs and 550 lncRNAs that regulate follicular development in various methods, many of which are involved in oocyte meiosis, progesterone-mediated oocyte maturation, and cell cycle [122]. As shown in Table 2.

CircRNAs are a novel type of noncoding RNA that form a covalently closed continuous loop, and they lack the 5′ terminal cap structure and 3’ terminal ploy (A) tail; these noncoding RNAs are ubiquitous in living organisms [128]. With the development of sequencing technology, some important circRNAs that participate in various biological functions have been discovered [20]; circRNA can be used as a biomarker of diseases to guide production practices [129]. Recent studies have revealed that circRNA molecules have numerous miRNA-binding sites; thus, they can function as miRNA sponges, as well as the regulators of splicing and transcription, and then indirectly regulate the expression of downstream target genes of miRNAs [130]. Studies on circRNA have focused on its role in only human and mouse granulosa cell development, and only a few studies have investigated the role of circRNAs in layer hen GCs, necessitating further studies. As shown in Table 3.

#### 4.3.2. The Role of DNA Methylation in Granulosa Cell Development

Environmental factors can impact epigenetic DNA modification by modifying the gene activity or regulating the gene expression. Epigenetic modification is a type of DNA variation that influences gene expression without altering the DNA coding sequence [137,138,139]. The DNA methylation level is mainly determined by the presence of CpG island in the promoter region. CpG differential methylated regions (DMRs) are significant epigenetic modification markers and functional areas involved in gene transcription [140,141,142]. To begin, 5 progesterone-mediated oocyte maturation genes, namely cell division cycle 27 (*CDC27*), adenylate cyclase 8 (*ADCY8*), AKT serine/threonine kinase 3 (*AKT3*), and microtubule-associated serine/threonine kinase 2 (*MAST2*), were discovered by the KEGG pathway enrichment analysis of the hypothalamus and ovary genes [143]. *CDC27* is a protein required for encoding cell cycle progression, and it is believed to be involved in cell proliferation and division [144,145]. The CpG island methylation level of this gene was reported to be high in Langshan chickens with a high degree of inbreeding, indicating that methylation of this gene affected transcription expression, thereby affecting oocyte maturation, which was one of the main reasons for the decline in reproductive performance of the inbred Langshan hens [146].

Furthermore, laying hens serve as a model for human ovarian cancer research [147]. In laying hens, DNA methyltransferases 1, 3A, 3B (*DNMT1*, *DNMT3A*, and *DNMT3B*) genes are strongly expressed in GCs of cancerous ovaries, and miR-1741, miR-16c, miR-222, or miR-1632 directly bind to the *DNMT3A* or *DNMT3B* transcripts. Following transcription, the expression of *DNMT3A* and *DNMT3B* genes is regulated. The expression level of DNMTs was determined in normal and cancerous ovaries, and the miRNA target validation assays indicated that the *DNMTs* are regulated by the post-transcriptional effects of particular miRNAs [148]. The *DNMT* genes exhibit cell-specific expression patterns, and the present review elaborates on the methylation state of the CpG island and the promoter region of a tumor suppressor gene in normal and cancerous laying hen ovaries.

*DNMT3A* and *DNMT3B* may methylate semi-methylated or unmethylated CpG islands at the same rate. Although the overall structure of *DNMT3* is comparable to that of DNMT1, its total length is shorter, and it carries a distinct proline–tryptophan–tryptophan–proline (*PWWP*) tetrapeptide. Furthermore, depending on the type of tumor, overexpression of *DNMT3A* or *DNMT3B* is linked to tumorigenesis in humans [149]. These findings imply that *DNMT3A* and *DNMT3B* function as DNA methyltransferases and play vital roles in normal development and diseases [150].As shown in Figure 4

## 5. Interaction Network Regulation Mechanism of the HPO Axis

### 5.1. Organism-Level Regulation

The HPO axis is crucial for improving egg production performance. From a systematic standpoint, clarifying the molecular mechanism of the HPO axis control of laying performance is crucial. The laying performance is directly regulated at the organism level and indirectly regulated by the ‘Gut–Brain Axis’. Organism-level regulation induces the release of signal molecules directly to the hypothalamus, triggering GnRH production, acting on the pituitary gland to secrete FSH and LH and regulating the target ovary tissues to release E_2_ and P_4_. These hormones regulate the proliferation, differentiation, and development of ovarian follicles at the organism level, thereby improving egg production performance. Simultaneously, dietary status and environmental factors directly or indirectly influence the HPO axis.

### 5.2. Cellular-Level Regulation

At the cellular level, the HPO axis is controlled by several processes. First, the liver cells produce and release IGF-I in response to GHs secreted by the pituicytes. Therefore, IGF-I is sometimes referred to as somatomedins (SM). Second, transforming growth factor and EGF indirectly regulate the synthesis and development of ECM and primary follicles. Third, certain reproductive functions are directly linked to the immune regulatory factors that regulate ovarian follicle formation through the ‘neuro–endocrine–immune’ pathway comprising T cells, TNF-α, and TNF-β produced by macrophages.

### 5.3. Molecular-Level Regulation

Many genes and signaling pathways jointly regulate the HPO axis and perform biological functions at the molecular level. First, egg production is determined by follicular selection, which facilitates the regulation of steroid-related genes *StAR* and *CYP11A1* that mediates the TGF-β signaling pathway. Second, *FSHR, OCLN, AMH*, and other genes in the Wnt signaling pathway, mTOR signaling pathway, and ECM signaling pathway influence follicular regeneration or delay follicular atresia by enhancing gene expression to accomplish follicular renewal and improve hens’ laying performance. Third, miRNA and lncRNA are also involved in the proliferation, differentiation, and atrestic regulation of the follicular GCs of hens. MiRNAs and lncRNAs may be involved in forming follicular cells, potentially providing a novel means and methodology for the molecular breeding of chickens. Fourth, DNA methylation can directly affect the methylation level of each tissue along with the HPO axis and significantly affect the methylation level of the CpG island in associated tissues. Furthermore, the liver connects hepatic lipid metabolism to the reproductive cycle through *ERα*, and the primary mechanism is related to yolk production through VLDL.

The laying performance is directly controlled by the HPO axis, thereby providing key genes and molecular signaling pathways involved in egg production, which can be identified through the genome, transcriptome, and other emerging technologies. Moreover, selecting key gene SNP markers can improve the efficiency of the choice of livestock and poultry breeds, laying a foundation for breeding new breeds.

## 6. Recommendations for Future Research

The economic benefits of large-scale chicken egg farms are determined by laying performance, which is one of the most important economic aspects of laying hen production. The HPO axis, which regulates egg production, is connected to the adrenal gland, thyroid gland, gut–brain axis, and liver lipid metabolism. Many researchers are investigating the differential expression genes and signaling pathway networks across the hypothalamus, pituitary, and ovarian tissues; however, from the perspective of system biology, studies investigating the key HPO axis controlling genes and signaling pathways of these function are limited. Therefore, future studies should focus on the following aspects:(1)Compared with the common transcriptome, single-cell RNA sequencing is a high-precision technique that can provide accurate resolution from the multicellular level to a single-cell level. This separates specific characteristics of the reaction cells or a group of cells based on RNA abundance to evaluate the condition of a single cell. Additionally, the single-cell RNA sequencing can be valuable for studying different cell types and developmental track. Therefore, single-cell RNA sequencing is a valuable tool for studying the molecular mechanism of GCs involved in the regulation of the egg production performance, which can effectively improve the efficiency of molecular breeding and accelerate breed selection.(2)Metabolomics is often referred to as the bridge between genomics and phenotype, and metabolic components can be used as biomarkers for determining complex biological traits. Using the transcriptome data to obtain a large number of differentially expressed genes and differential metabolites obtained through correlation analysis, and from two levels, causes and consequences of organism are analyzed, identifying key gene targets, metabolites, and metabolic pathways to build the key control networks. Efficient seed selection based on biomarkers is also the development direction in the breeding of layer hens.(3)Proliferation, differentiation, and apoptosis of GCs are the key to prolonging the laying cycle, and GC gene expression is temporally and spatially regulated. Future studies should utilize single-cell RNA sequencing and spatial transcriptomics to analyze the gene expression from both temporal and spatial dimensions in GCs. Moreover, the chromatin accessibility analysis technique can be used to predict the dynamic changes of chromatin conformation during follicle development, which may provide ideas for the breeding of high-reproductive performance hens.

## Figures and Tables

**Figure 1 genes-14-00141-f001:**
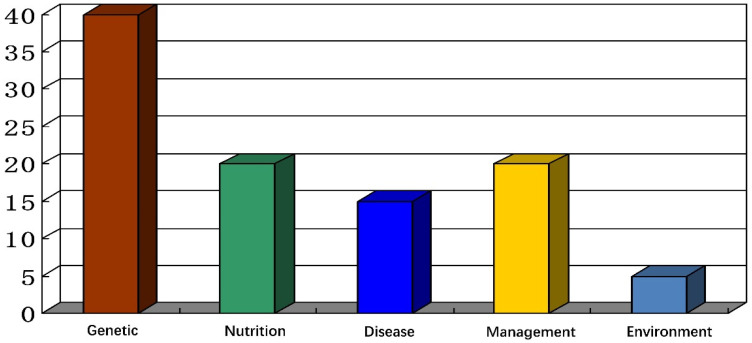
Factors influencing layer hens egg production over the last 50 years.

**Figure 2 genes-14-00141-f002:**
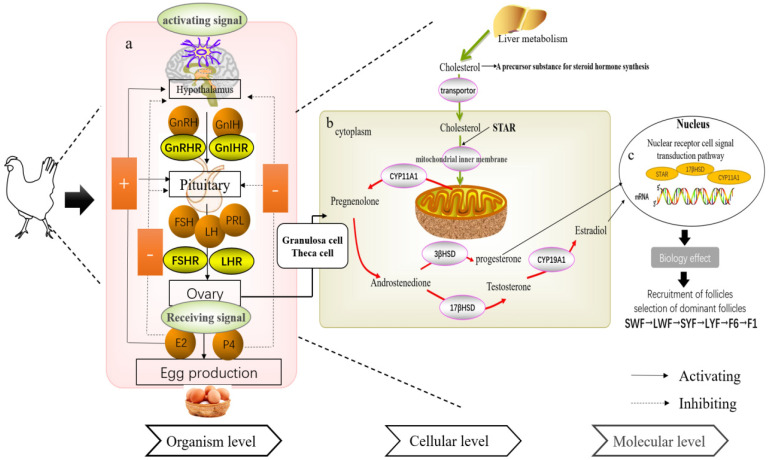
(**a**–**c**) Interaction mechanism of the avian HPO axis network.

**Figure 3 genes-14-00141-f003:**
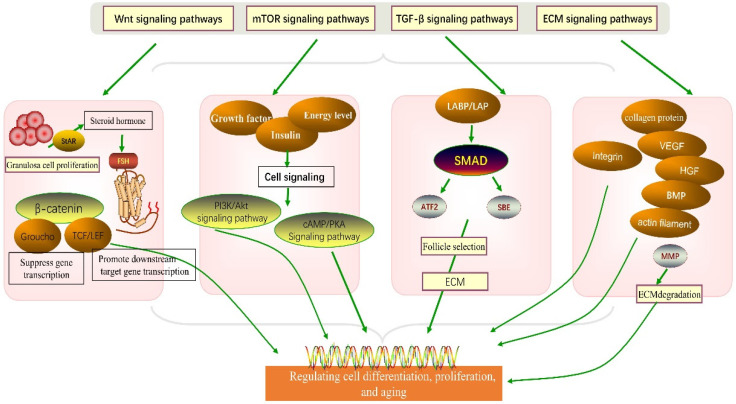
The role of four signaling pathways in granulosa cell development.

**Figure 4 genes-14-00141-f004:**
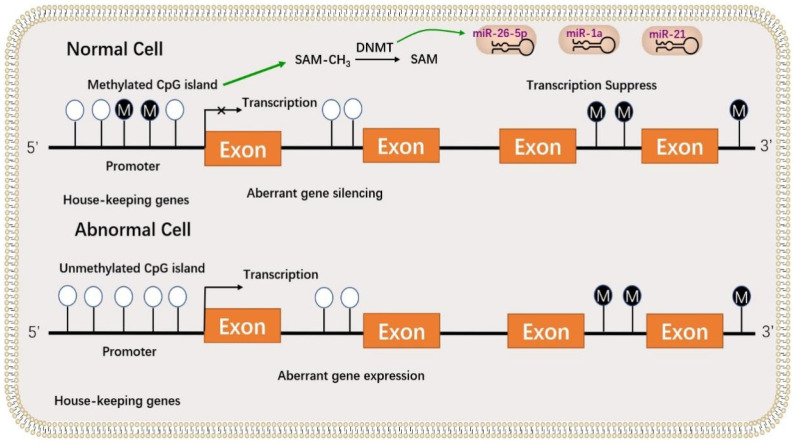
The role of DNA methylation in granulosa cell development. There are two main patterns of DNA methylation, including methylation of CpG islands (normal cells) and unmethylation of CpG islands (abnormal cells). These CpG islands are located in the promoter regions of housekeeping genes. If the CpG islands are methylated in the promoter region, the expression of downstream genes is inhibited, and cell development is normal. If the CpG islands are unmethylated in the promoter region, downstream genes are activated, and cell development is abnormal. The CpG island for methylation donors complete the process of demethylation under the action of the DNMT enzyme, which combines with miRNAs (miR-26-5p, miR-1a, and miR-21) to suppress transcription.

**Table 1 genes-14-00141-t001:** Species and role of miRNA in granulosa cell development.

miRNA	Target Genes	Signaling Pathways	Function	Tissue	Reference
miR-29c-3p	*FOXL2*	cAMP/focal adhesion	Transcription factor Promotes differentiation of pre-hierarchical GCs (PhGCs) and preovulatory GCs (PoGCs)	Chicken (PhGCs/PoGCs)	[71]
/	*WNT6*/*LRP1*/*FOXO1*	Wnt/β-catenin dependent pathway	WNT6 cooperates with FSH to promote follicle development	Hyline-brown (SYF)	[103]
gga-miR-30a-5p	*BCL1*	miR-30a-5p regulatory pathway	Activates (proliferation)	Chicken	[104]
gga-miR-146b-3p	*AKT1*	PI3K/ATK	Inhibits (proliferation)	Chicken	[105]
miR-458b-5p	*CTNNB1* (3′UTR)	Wnt/β-catenin	Inhibits (proliferation)	Chicken (Hyline Brown )	[106]
miR-26a-5p	*TNRC6A*	/	facilitates chicken ovarian cell proliferation	Chicken (ovarian)	[100]
miR-23b-3p	*GDF9*	TGF-β signaling pathway	Inhibits (proliferation) steroid hormone synthesis	Chicken (SYF)	[107]
miR-200a	*ZEB1*/*SIP1*/*SIRT1*	reproduction regulation pathways	Inhibit granulosa cells proliferation	Lu hua chicken	[10]
miR-138-1-3p	*COL4A5*	matrix metalloproteinase (MMP)	Remodeling of the extracellular matrix during ovarian follicle development in chickens	Chicken (largest preovulatory follicles)	[108]
miR-449b-5p	*IGF2BP3*	Steroid hormone synthesis signaling pathway	Regulate the expression of key steroidogenesis-related genes (StAR and CYP19A1)	Chicken (SYF)	[109]
miR-135a-5p	*KLF4*/*ATP8A1*/*CPLX1*	p53 Signaling pathway	Involve in proliferation and differentiation in chicken ovarian follicular	Chicken	[110]
miR-138-5p	*SIRT1*	Apoptosis signaling pathway	Promotes apoptosis and follicular atresia	Chicken	[111]
miR-10a-5p	*MAPRE1*	/	Inhibits proliferation and progesterone synthesis	Tian fu chicken	[112]
miR-122-5p	*MAPK3*	/	Promotes apoptosis through the post-transcriptional downregulation of MAPK3.	Chicken (hierarchal follicles)	[113]
miR-302a-3p	*DRD1*	/	Inhibits GCs proliferation	Chicken (Small yellow follicle)	[114]

Abbreviations: Forkheadbox L2 (FOXL2); WNT family member 6 (Wnt6); forkhead box transcription factor O1 (FOXO1); B-cell lymphoma 1 (BCL1); trinucleotide repeat-containing gene 6a (TNRC6a); growth and differentiation factor 9 (GDF9); collagen type IV α 5 chain (COL4A5); insulin-like growth factor 2 mRNA-binding protein 3 (IGF2BP3); dopamine receptors 1 (DRD1); Krüppel-like factor 4 (KLF4); ATPase phospholipid transporting 8A1 (ATP8A1); complexin-1 (CPLX1); silencing information regulator 2 related enzyme 1 (SIRT1); mitogen-activated protein kinase 3 (MAPK3); microtubule associated protein RP/EB family member 1 (MAPRE1); catenin β 1 (CTNNB1); zine finger E-box binding homeobox (ZEB1); Smad interacting protein 1 (SIP1).

**Table 2 genes-14-00141-t002:** Species and role of lncRNAs in granulosa cell development.

LncRNA	Target Gene	Function	Species	Reference
LncRNAXLOC_110025	*MOS BMP15 WNT6 CDC25A*	Promote follicle development	Hy-line Brown	[122]
LncRNA_138134	*PSMD6*	Affect oocyte maturation	Human (GCs)	[123]
LncRNA_210520.2	*SOWAHA*	Inhibited the proliferation of granulosa cells	Chicken (SYF)	[124]
Lnc RNAGLM	*ERα*	Regulated by estrogen through ERα	Laying hens	[125]
LncRNALTR	*ERβ*	Involved in lipid metabolism	Chicken	[10]
LncRAN MTSRG.17017.1 MTSRG.6475.20	*CACNA1C* *TGFB1*	Affect gonad development and GnRH signaling pathway	White Leghorns and Beijing You chickens	[126]
LncRNA XLOC_001347 XLOC_016063 XLOC_02660 XLOC_03201 XLOC_005141	*CASP6*/*MMP2*/*SMAD2*	Involved cell growth, proliferation, and development	Broody chickens (BC) and normal ovaries (NO)	[52]
LncRNA RP4-545C24.1	*RAD51 WT1*	Inhibition of DNA damage repair capacity	Human	[127]

Abbreviations: MOS (MOS proto-oncogene, serine/threonine kinase); PSMD6 (proteasome 26S subunit, non-ATPase 6); sosondowah ankyrin repeat domain family member A (SOWAHA); calcium volt aggregated subunit alphal C (CACNA1C); transforming growth factor beta1 (TGFB1); caspase 6 (CASP6); RAD51 (RAD51 recombinase); WT1 transcription factor (WT1).

**Table 3 genes-14-00141-t003:** Species and role of circRNAs in granulosa cell development.

CircRNA	miRNA	Target Gene	Function	Species	Reference
circRNA-aplacirc_13267	apla-miR-1-13	*THBS1*	Promotes granulosa cell apoptosis	Duck	[131]
circRNA_RANBP9	miR-136-5p	*XIAP*	Inhibits granulosa cell apoptosis	Human (Polycystic ovary syndrome)	[132]
circRNA_EML1	miR-449a	*IGF2BP3*	Promotes granulosa cell steroid hormone synthesis and estrogen and progesterone secretion, down regulate miR-449a	Hyline Brown	[133]
circRNA_0320 circRNA_0185	miR-143-3p	*FSHR*	Promotes GC differentiation and follicle development	Chicken (SYF)	[134]
sss-circINHA-001	miR-24-5p miR-7144-3p miR-9830-5p	*INHA*	Resisting GC apoptosis and follicular atresia	Chicken	[135]
circRNA_8:6369673|6402248 circRNA_8:6369673|642209 circRNA_8:6384248|6402248	miR-1625-3p miR-1552-3p miR-16-2-3p miR-18b-3p miR-200a-3p	*RalGPS2*	Regulate GC development	Chicken (SYF/F6/F1)	[72]
novel_circ0004730	/	*ESR*	Cell growth, proliferation, differentiation, and apoptosis	Chicken (SYF/F6/F1)	[136]

Abbreviations: throbospondin1 (THBS1); X-linked inhibitor of apoptosis (XIAP); RalGEF with PH domain and SH3 binding motif 2 (RalGPS2).

## Data Availability

The data that support this study are available in the article.

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
