# Peer review of "Interacting Networks of the Hypothalamic–Pituitary–Ovarian Axis Regulate Layer Hens Performance"

_genes, 2023, doi:10.3390/genes14010141_

Round 1

Reviewer 1 Report (New Reviewer)

The authors have attempted to provide a review on the role of Hypothalamic–Pituitary–Ovarian Axis in regulating layer hen performance.

The manuscript provides a lot of information, but its very in cohesive and jumps from paragraph to paragraph and in multiple places there is no connection between paragraphs. I suggest that the authors edit and rephrase parts of the manuscript to make it a cohesive read. Instead of attempting to explain everything that is known about HPO axis and its role in regulating laying performance, focus of a few critical things and explain it well.

Since this manuscript is a review, everything explained here-in should be in third person.

For example "Transcriptomics can help identify differentially expressed genes in the hypothalamus" on line 157, should be rephrased as DEGs were identified in the hypothalamus and pitutary tissues of high and low yield hens". These kind of changes must be made at multiple places in the manuscript.

Minor Points :

Abstract :- line 19-20 'regulation of the HPO axis in the laying performance of layer hens at the organism', consider changing it to 'The current review elaborates on the regulation of the HPO axis and its role in the laying performance of hens at the organism, cellular, and molecular levels'

line 39 citation should be 2 and not 142. Correct and update citation order.

 Gene names throughout the manuscript must be in italic.

Section 2.2.1 is confusing and must be rewritten.

Citation must be provided at the first instance a study is referred to. for example

on line193 'cultured hepatocytes of adult rats', citation for lines 223-226 etc. Similar corrections should be made throughout the manuscript.

Author Response

Response to Reviewer 1 Comments

Thank you for your letter and for the reviewers' comments concerning our manuscript entitled “Interacting Networks of the Hypothalamic–Pituitary–Ovarian Axis Regulate Layer Hen Performance”(ID: genes-2073765), and we have done our best to revise them according to the requirements. And the point-to-point modifications are shown below. See Tracks Changes and blue in the manuscript for details of the revisions

Point 1: The manuscript provides a lot of information, but its very in cohesive and jumps from paragraph to paragraph and in multiple places there is no connection between paragraphs. I suggest ·that the authors edit and rephrase parts of the manuscript to make it a cohesive read. Instead of attempting to explain everything that is known about HPO axis and its role in regulating laying performance, focus of a few critical things and explain it well.

Response 1: thanks.Your comments is helpful for me. I added some connectives in some chapters of the article and modified some contents to ensure the continuity of the article content.Such as‘Furthermore’、‘Research shown that……’. ‘these result indicated that……’ In this way, there will be a link between the paragraphs, reflecting the system and continuity of the article.

Point 2: Since this manuscript is a review, everything explained here-in should be in third person.

For example "Transcriptomics can help identify differentially expressed genes in the hypothalamus" on line 157, should be rephrased as DEGs were identified in the hypothalamus and pitutary tissues of high and low yield hens". These kind of changes must be made at multiple places in the manuscript. 

Response 2: thanks.Your comments is helpful for me.we have modifed it,At the same time, I also made a comprehensive revision to other similar parts of the paper.

Point 3: Abstract :- line 19-20 'regulation of the HPO axis in the laying performance of layer hens at the organism', consider changing it to 'The current review elaborates on the regulation of the HPO axis and its role in the laying performance of hens at the organism, cellular, and molecular levels'. 

Response 3: thanks.Your comments is helpful for me. I have changed it in the article. See the part of the manuscript marked in yellow and Traces Changes for detailed revisions

Point 4: line 39 citation should be 2 and not 142. Correct and update citation order. 

Response4: thanks. Your ideas are very helpful to improve the quality of my article. I have adjusted it according to the literature order.

Point 5: Gene names throughout the manuscript must be in italic.

Response 5: thanks.we have modified,We have completed the full text of the required italicized gene names, Meanwhile, I have attached the full names of the genes that first appeared in the paper to the bottom of the table

Point 6: Section 2.2.1 is confusing and must be rewritten.

Response 6: thanks.Your comment is helpful for me. I have also carefully reviewed the latest relevant literature and reprocessed this paragraph.I have rewritten,please see:

Nutrition affects the secretion of reproductive hormones and the concentration of metabolic substrates in blood. Reasearch shown that food restriction promoted released of P4,reduced out of the T,but did not affect E release, suggesting that metabolic status can control the release of these hormones[152]. The estrogen estradiol-17ß is known as one of the major gonadal steroid hormones with different functions in reproduction[153],Morden layer hens have consistently high plasma estradiol-17ß levels[154],estrogen divided into two types recptor, ER receptor α and ER receptor β,the mRNA expression of Erα was significance higher compared to ERβ in layer hens granulosa cells[155],hence,It suggested that estrogen plays a role in regulating follicular development through ERα receptors. In previous research reported that follicle selection is indispensable in the reproductive process in female chickens, the symbol of which the proliferation and differentiation of granulosa cells(GCs), In chickens follicular GCs, the expression of steroidogenic acute regulatory protein (StAR) and cytochrome P450 family 11 subfamily A member 1 (CYP11A1) is a prerequisite for progesterone synthesis, and is related to follicle selection[156-158].It suggested that estrogen and progesterone work together to complete the physiological process of follicle selection in layer hens. Many studies have shown that estrogen is critical for lipid metabolism in the liver, In vivo and in vitro experiments showed that estrogen decreased the mRNA expression of PPARGC1B, which had been reported to be linked with lipid metabolism, by directly increasing the expression of miR-144-3p[159][160].these the findings of result shown that the liver ERα controls the regulation of liver lipid metabolism, thus connecting liver lipid metabolism to the reproduction cycle [30].

Point 7: Citation must be provided at the first instance a study is referred to. for example

on line193 'cultured hepatocytes of adult rats', citation for lines 223-226 etc. Similar corrections should be made throughout the manuscript. 

Response 7: thanks.Your comment is helpful for me. I have attached similar references to some parts of the article

Reviewer 2 Report (New Reviewer)

The manuscript "Interacting Networks of the Hypothalamic–Pituitary–Ovarian Axis Regulate Layer Hen Performance" submitted for review is an interesting review of the current data in the literature.

Both the construction of the description and the form of presentation do not raise objections, they deserve more praise. No excessive number of self-citations found.

However, the quality of figure 3 raises reservations. Unfortunately, in a complex study, the resolution of the image is so small that the letters seem blurred and it is difficult to read the content.

I believe that after improving the resolution of figure 3, the article is suitable for printing.

Author Response

Response to Reviewer 2 Comments

Thank you for your letter and for the reviewers' comments concerning our manuscript entitled “Interacting Networks of the Hypothalamic–Pituitary–Ovarian Axis Regulate Layer Hen Performance”(ID: genes-2073765), and we have done our best to revise them according to the requirements. And the point-to-point modifications are shown below. The revisions in the manuscript have been marked in blue and Tracks Changes

Point 1:  the quality of figure 3 raises reservations. Unfortunately, in a complex study, the resolution of the image is so small that the letters seem blurred and it is difficult to read the content.

Response 1: thanks.Your comment is right. We have changed the picture to improve the quality and clarity of figure 3.

Reviewer 3 Report (New Reviewer)

Which is better, "layer hens" or "laying hens"? Prease reconsider.

I would like to propose network analysis listed in Tables 1, 2, and 3 using public database such as STRING. This might provide more detailed information on the mechanism.

The orders of the references must be reconsider. For example, the reference in L39 should be changed from [142] to [2]. 

Author Response

Response to Reviewer 3 Comments

Thank you for your letter and for the reviewers' comments concerning our manuscript entitled “Interacting Networks of the Hypothalamic–Pituitary–Ovarian Axis Regulate Layer Hen Performance”(ID: genes-2073765), and we have done our best to revise them according to the requirements. And the point-to-point modifications are shown below. See the blue marks and Tracks Changes  in the manuscript for details of the revisions

Point 1: Which is better, "layer hens" or "laying hens"? Prease reconsider. 

Response 1: thanks.Your comments is helpful for me. I think ‘layer hens’ is right, it is more in line with what the article is trying to say. I have changed it in the article. See the part of the manuscript marked in yellow and Traces Changes for detailed revisions

Point 2: I would like to propose network analysis listed in Tables 1, 2, and 3 using public database such as STRING. This might provide more detailed information on the mechanism.

Response 2: Thanks. Your ideas are very helpful to improve the quality of my article. However,Most studies on non-coding RNA focus on miRNA, but all ncRNA have been not studied, and it is difficult to find common ground to construct the mechanism network map. I also tried and failed to build a network diagram using STRING databases,We believe that non-coding RNA can be further studied in the future, I will also further learn to use this databases, especially your advice and guidance.

Point 3: The orders of the references must be reconsider. For example, the reference in L39 should be changed from [142] to [2].

Response 3: thanks.we have modified,This paper has been modified according to the literature arrangement order of Genes magazine.

Round 2

Reviewer 1 Report (New Reviewer)

Thank you for responding to my comments and making the changes suggested.

Reviewer 3 Report (New Reviewer)

The manuscipt has been modified.

This manuscript is a resubmission of an earlier submission. The following is a list of the peer review reports and author responses from that submission.

Round 1

Reviewer 1 Report

This submission is designed to discuss the development of the ova re the hypothalamic-pituitary-ovarian axis.  It does address follicular development, but not the formation of the egg and oviposition except in general terms relating to improving egg production.  It is a long stretch from follicular development to oviposition and an inference of addressing peak egg production and persistency of lay in commercial poultry.

The review goes beyond its title when it naively generalizes on application.  The authors may be wise to stay within the area of their expertise.   

References need editing for consistency.  There are many references in this manuscript, but often it is unclear from the text and reference section when citied. 

l. 18. "in hens" - the title says in hens - males don't lay eggs.

l. 37-39 and Figure 1. Do you have a reference for the %?

l. 44. Are you including turkeys in this review (reference 7)?  Not clear as you refer to laying hen industry (l. 29), which set my thinking for egg-type chickens not broilers or turkeys.  Would be helpful to define what is addressed.  The abstract had me focusing on egg stocks re line 20.  Essentially what you wrote could apply to all avian species.

l. 79. gene vs genes?  My sense is that it is genes.  To assume otherwise is naive.

l. 92. "each day" - is this restriction necessary rather than a "sequential order"? 

l. 84-85. It is not clear where statements are made and where the references belong.

It is not clear for the figures if they are yours or from a reference eg. Fig 1, 2...

l. 147-154. Need references.

Section 2.2.3 Genetic factors is wanting.  What are the quantitative aspects per onset, intensity, and persistency of lay?  Last sentence (l. 172-174) lost me as to what is a "new" phenotype.

Sections 5.2 and 5.3 lack any citation.  Why?

l. 516-517. "which may provide...human reproductive diseases." This phrase sums up where the authors go far beyond the material they reviewed.

Author Response

Response to Reviewer 1 Comments

Thank you for your letter and for the reviewers' comments concerning our manuscript entitled “Interacting Networks of the Hypothalamic–Pituitary–Ovarian Axis Regulate Layer Hen Performance”(ID: genes-1937245), and we have done our best to revise them according to the requirements. And the point-to-point modifications are shown below. See the yellow marks in the manuscript for details of the revisions

Point 1:The review goes beyond its title when it naively generalizes on application.  The authors may be wise to stay within the area of their expertise.   

Response 1: thanks.Your comments is helpful for me. Firstly There are some cases of pathology in the manuscript, however, these cases are aimed at confirming that pathways or genes do play a role in layer hen peformance. In the meantime, we have replaced some of the references with the latest references to chickens. See the part of the manuscript marked in yellow for detailed revisions

Point 2: References need editing for consistency. There are many references in this manuscript, but often it is unclear from the text and reference section when citied. 

Response 2: We have added references,

Point 3: l. 18. "in hens" - the title says in hens - males don't lay eggs.

Response 3: thanks.we have modified,add “layer”hens

Point 4: l. 37-39 and Figure 1. Do you have a reference for the %?

Response 4: thanks. We have add“[142]”Reference.

Point 5: l. l. 44. Are you including turkeys in this review (reference 7)?  Not clear as you refer to laying hen industry (l. 29), which set my thinking for egg-type chickens not broilers or turkeys.  Would be helpful to define what is addressed.  The abstract had me focusing on egg stocks re line 20.  Essentially what you wrote could apply to all avian species.

Response 5: thanks. Your comment is right.we have replaced it with a reference for layer hens

Point 6: l. 79. gene vs genes?  My sense is that it is genes.  To assume otherwise is naive.

Response 6: thanks. Your comment is right.we have modified,add”genes”

Point 7: l. 92. "each day" - is this restriction necessary rather than a "sequential order"? 

Response 7: thanks. Your comment is right.we have add”sequential order”

Point 8: l. 84-85. It is not clear where statements are made and where the references belong.

Response 8: thanks. Your comment is right. We have changed the relevant references, and the two stages of follicles are also mentioned in the literature.

Point9: l. 147-154. Need references.

Response 9: thanks. Your comment is right. We have added references

Point10: Section 2.2.3 Genetic factors is wanting.  What are the quantitative aspects per onset, intensity, and persistency of lay?  Last sentence (l. 172-174) lost me as to what is a "new" phenotype.

Response 10: Thanks,Your comment is right.Firstly, we performed a genome-wide association study (GWAS) in a mixed linear model. SNPS related to egg production were mined using GWAS.Secondly, we have changed “new phenotype”into”new breeds”

Point11:Sections 5.2 and 5.3 lack any citation.  Why?

Response 11: Thanks,Because these two chapters belong to the concluding chapters, they are not quoted     

Point12: l. 516-517. "which may provide...human reproductive diseases." This phrase sums up where the authors go far beyond the material they reviewed.

Response 12: Thanks, Your comment is right. We have deleted the relevant expression

Reviewer 2 Report

Sir,

    The organization of the manuscript is good. The topic discussed in current context is quite relevant and of practical importance. I have two major concerns:

1. Firstly,the topic is focussed on egg production of hens, and factors affecting it, including molecular and genomic control. But, I observed a majority of the references from other species as bovines, goats, porcine and humans, even drosophila.

2. Secondly, The authors have discussed mostly the physiological roles for ‘hypothalamic–pituitary–ovarian (HPO) axis’ determining the egg production rate. However the references include a lot for pathological cases, e.g.ovarian Neoplasms, polycystic ovary syndrome, and ovarian aging., cancer etc.

As per my suggestion, a separate section on the molecular study for pathogen changes may be studied.

Thanking you.

Author Response

Response to Reviewer 2 Comments

Thank you for your letter and for the reviewers' comments concerning our manuscript entitled “Interacting Networks of the Hypothalamic–Pituitary–Ovarian Axis Regulate Layer Hen Performance”(ID: genes-1937245), and we have done our best to revise them according to the requirements. And the point-to-point modifications are shown below. The revisions in the manuscript have been marked in yellow

Point 1: 1. Firstly,the topic is focussed on egg production of hens, and factors affecting it, including molecular and genomic control. But, I observed a majority of the references from other species as bovines, goats, porcine and humans, even drosophila.

Response 1: thanks.Your comment is right. We've replaced the literature on chickens, See the yellow marks in the manuscript for details of the revisions

Point 2: 2. Secondly, The authors have discussed mostly the physiological roles for ‘hypothalamic–pituitary–ovarian (HPO) axis’ determining the egg production rate. However the references include a lot for pathological cases, e.g.ovarian Neoplasms, polycystic ovary syndrome, and ovarian aging., cancer etc.

Response 2: thanks. Your comments are very helpful to me, Many pathological cases are included in my manuscript, mainly because laying hens are the model animals for human ovarian cancer research, and these contents are also included in the literature retrieved

Reviewer 3 Report

The title and abstract of this review article suggested that the manuscript would be discussing the HPO axis and layer hen performance. However, the vast majority of references used were bovine, ovine, feline, rat and human, without any clarification in the text. In some sections, the reader was led to believe that these were references to avian research. 

In section 2, the description of the HPO axis had many incorrect references and statements. For example, in birds, GnRH-I is the primary gonadotropin responsible for the activation of the axis. GnRH-II is said to control behaviours instead. Also, the references used on line 72 are again missing any avian information. In line 65-68, there are misleading statements. For example, E2 peaks at the time of the first egg, not during the late stages of lay. 

The information on leptin in section 2.2.1 is extremely misleading. Leptin was discovered in birds in 2016 (Seroussi et al., 2016), so any research prior to this time must be discussed at length for its inconsistencies. The role of leptin has been largely debated in the chicken and it is not as simple as outlined. What is outlined by the author's is more consistent with mammals. This entire section would require a re-write or removal. 

For the remainder of the manuscript, there is a large absence of new data presented in avian species. Much of the later sections use only mammalian references and fail to state this or make hypotheses reagrding birds. This is unfortunate as many of the topics discuss do in fact have relevant recent research available.

Author Response

Response to Reviewer 3 Comments

Thank you for your letter and for the reviewers' comments concerning our manuscript entitled “Interacting Networks of the Hypothalamic–Pituitary–Ovarian Axis Regulate Layer Hen Performance”(ID: genes-1937245), and we have done our best to revise them according to the requirements. And the point-to-point modifications are shown below. Changes in the manuscript are marked in yellow

Point 1: The title and abstract of this review article suggested that the manuscript would be discussing the HPO axis and layer hen performance. However, the vast majority of references used were bovine, ovine, feline, rat and human, without any clarification in the text. In some sections, the reader was led to believe that these were references to avian research. 

Response 1: thanks. Your comments are very helpful to me.we have replaced the chicken reference as requested.

Point 2: In section 2, the description of the HPO axis had many incorrect references and statements. For example, in birds, GnRH-I is the primary gonadotropin responsible for the activation of the axis. GnRH-II is said to control behaviours instead. Also, the references used on line 72 are again missing any avian information. In line 65-68, there are misleading statements. For example, E2 peaks at the time of the first egg, not during the late stages of lay. 

Response 2: thanks. Your comments are very helpful to me. Firstly,In line 65-68,we have transform into “at the time of the first egg”  Secondly, Through literature review, it is found that GnRH-Ⅱ plays the main role in chicken,please see above:

Gonadotropin-releasing hormone (GnRH) plays a pivotal role in the physiology of reproduction in mammals. GnRH acts by binding to the GnRH receptor (GnRHR). In humans, only 1 conventional GnRH receptor subtype (type I GnRH receptor) has been found. In the human genome, 2 forms of GnRH have been identified, GnRH-I (mammal GnRH) and GnRH-II (chicken GnRH II). 

Metallinou C, Asimakopoulos B, Schröer A, Nikolettos N. Gonadotropin-releasing hormone in the ovary. Reprod Sci. 2007 Dec;14(8):737-49. doi: 10.1177/1933719107310707. PMID: 18089592.

Point 3: The information on leptin in section 2.2.1 is extremely misleading. Leptin was discovered in birds in 2016 (Seroussi et al., 2016), so any research prior to this time must be discussed at length for its inconsistencies. The role of leptin has been largely debated in the chicken and it is not as simple as outlined. What is outlined by the author's is more consistent with mammals. This entire section would require a re-write or removal. 

Response 3: thanks. Your comment is right. We have removed the reference to leptin

Point 4: For the remainder of the manuscript, there is a large absence of new data presented in avian species. Much of the later sections use only mammalian references and fail to state this or make hypotheses reagrding birds. This is unfortunate as many of the topics discuss do in fact have relevant recent research available.

Response 4: thanks. Your comment is right. We have added a large amount of up-to-date literature on chickens, Please refer to the revised draft for details